# Prediction Model of Hemorrhage Transformation in Patient with Acute Ischemic Stroke Based on Multiparametric MRI Radiomics and Machine Learning

**DOI:** 10.3390/brainsci12070858

**Published:** 2022-06-29

**Authors:** Yucong Meng, Haoran Wang, Chuanfu Wu, Xiaoyu Liu, Linhao Qu, Yonghong Shi

**Affiliations:** 1Digital Medical Research Center, School of Basic Medical Sciences, Fudan University, Shanghai 200032, China; ycmeng21@m.fudan.edu.cn (Y.M.); hrwang20@fudan.edu.cn (H.W.); cfwu14@fudan.edu.cn (C.W.); liuxiaoyu21@m.fudan.edu.cn (X.L.); lhqu20@fudan.edu.cn (L.Q.); 2Shanghai Key Laboratory of Medical Imaging Computing and Computer Assisted Intervention, Shanghai 200032, China

**Keywords:** radiomics, hemorrhagic transformation, acute ischemic stroke, machine learning

## Abstract

Intravenous thrombolysis is the most commonly used drug therapy for patients with acute ischemic stroke, which is often accompanied by complications of intracerebral hemorrhage transformation (HT). This study proposed to build a reliable model for pretreatment prediction of HT. Specifically, 5400 radiomics features were extracted from 20 regions of interest (ROIs) of multiparametric MRI images of 71 patients. Furthermore, a minimal set of all-relevant features were selected by LASSO from all ROIs and used to build a radiomics model through the random forest (RF). To explore the significance of normal ROIs, we built a model only based on abnormal ROIs. In addition, a model combining clinical factors and radiomics features was further built. Finally, the models were tested on an independent validation cohort. The radiomics model with 14 All-ROIs features achieved pretreatment prediction of HT (AUC = 0.871, accuracy = 0.848), which significantly outperformed the model with only 14 Abnormal-ROIs features (AUC = 0.831, accuracy = 0.818). Besides, combining clinical factors with radiomics features further benefited the prediction performance (AUC = 0.911, accuracy = 0.894). So, we think that the combined model can greatly assist doctors in diagnosis. Furthermore, we find that even if there were no lesions in the normal ROIs, they also provide characteristic information for the prediction of HT.

## 1. Introduction

Cerebrovascular disease is one of the three major diseases that endanger human health. In cerebrovascular disease, ischemic cerebrovascular diseases have high incidence rate, high disability rate, and high mortality, which are seriously harmful to human health. For this disease, the saying “time is brain” is true as timely, decisive, and effective treatment is essential to the prognosis of patients.

Intravenous thrombolysis with recombinant tissue plasminogen activator (rt-PA) is the most common drug therapy for patients with acute ischemic stroke. Studies have shown that intravenous thrombolysis with rt-PA within 4.5 h of the onset of the disease can open the occluded blood vessels of the patient as soon as possible, restore the blood perfusion of the cerebral tissue, and save the ischemic penumbra tissue, improve the prognosis and reduce the disability rate [1]. However, this therapy is often accompanied by the complication of HT. Once HT occurs, the disability rate and mortality rate of patients can be as high as 90.0% and 50.0–80.0%, respectively [2,3]. Considering the risk of HT, doctors cannot make decisions quickly, which affects the rescue of patients, and severely limits the clinical application of rt-PA intravenous thrombolytic therapy.

Early studies aimed to predict the risk of HT by comprehensively evaluating the clinical indicators of patients [4,5,6]. However, this method requires doctors to fully understand various clinical indicators of patients, and its prediction accuracy mainly relied on the doctor’s experience. Therefore, the use of rt-PA in patients cannot be well guided.

In recent years, many scholars have successively proposed that, compared with clinical information, cerebral perfusion images with different MRI parameters can more intuitively reflect the patient’s brain tissue ischemia and the degree of blood–brain barrier (BBB) damage, so they have more reliable HT prediction performance. Ueda et al. [7] used Single-Photon Emission Computed Tomography (SPECT) to confirm that a decrease in cerebral blood flow (CBF) before arterial thrombolysis was associated with HT, Alsop et al. [8] found that cerebral blood volume (CBV) after thrombolysis was better than apparent diffusion coefficient (ADC) in predicting the occurrence of HT, with a sensitivity of 1.0 and a specificity of 0.73, Scalzo et al. [9] pointed out that the absence of time-to-peak (TTP) image can clearly distinguish high-risk HT groups and low-risk HT groups, and its sensitivity and specificity for predicting HT can be as high as 1.0 and 0.824, respectively. To sum up, local cerebral hypoperfusion in patients with ischemic stroke leads to the destruction of the integrity of BBB, and vascular recanalization after thrombolysis makes the blood components enter the brain parenchyma from the damaged BBB, which is the main reason for the risk of HT [10]. Perfusion imaging has more advantages in the detection of HT high-risk populations by comparison with non-contrast CT (NCCT). However, most of the predictions of HT by perfusion imaging are retrospective, and the sensitivity and specificity of each perfusion parameter need to be confirmed by further study.

Since 2012, Radiomics has been widely used in tumor detection [11,12,13], differential diagnosis [14,15], pathological typing and grading [16,17,18], and other fields. At the same time, Machine learning (ML), which discovers hidden rules from a large number of known data and applies them to the prediction and classification of unknown data, has been widely used in medical analysis and disease prediction [19,20]. For this reason, many studies began to combine radiomics with ML to recognize the risk of HT in stroke patients from the multiparametric MRI images: Bentley et al. [21] used NCCT and NIH Stroke Scale (NIHSS) to obtain the HT prediction accuracy of AUC = 0.74, Yu et al. [22] used different ML models to predict HT from source perfusion formaldehyde imaging data and obtained a classification accuracy of 0.88.

Besides, deep learning (DL) has recently emerged as a powerful tool for medical image analysis. As for HT prediction, Jiang et al. [23] built a DL model based on multiparametric MRI images by using diffusion-weighted imaging (DWI), mean transit time (MTT) and TTP sequences combined with clinical data. They first extracted the deep features of each slice using inception V3, then fused the features of all slices of each sequence, and finally spliced the features of all sequences with clinical features to realize HT prediction. Yu et al. [24], used long short-term memory (LSTM) for feature extraction and fusion of local patches in a perfusion-weighted imaging (PWI) sequence, and combined them with the corresponding features of DWI images to predict HT. Although the DL method has been proved to have great application potential in segmentation, classification, and other fields, they often need a large amount of data for model training. When the amount of data is not large enough, DL often leads to overfitting. In addition, the DL method has poor interpretability, which limits its wide application in clinical practice.

In this study, a reliable model for the prediction of HT is built to quickly evaluate whether patients can receive intravenous thrombolytic therapy. Firstly, radiomics features are extracted from both normal and abnormal ROIs of multiparametric brain MRI images, abnormal ROIs are in the hemisphere of the brain where hemorrhage occurred, while normal ROIs are in the healthy hemisphere of the brain. The LASSO regression [25] is used for feature selection, then the HT prediction model is built through RF based on the selected All-ROIs features. In addition, by removing features from the normal ROIs, an abnormal-ROIs HT prediction model is further built to evaluate the importance of the features from normal ROIs. Finally, the most distinctive clinical features are screened out and fused with radiomics ones, and an HT prediction model based on fusion features is constructed. We conducted experiments on real clinical data, the results show that the classification accuracy of the model based on All-ROIs is significantly higher than the method focusing only on the abnormal ROIs, and can be further improved by adding the important clinical information.

Our contributions are two-fold: First, we built a reliable model for pretreatment prediction of HT in patients with acute ischemic stroke by combing clinical factors and multiparametric MRI radiomics features. Second, as far as we know, the existing radiomics-based methods for HT prediction only focus on the abnormal ROIs, ignoring the relationship between abnormal and normal areas. We successfully noticed this and made use of normal-ROIs information to evaluate its importance for the first time. The experimental results show that the proposed HT prediction model is more effective.

## 2. Materials

### 2.1. Patients Enrollment

A total of 136 patients with acute ischemic stroke were collected from the Stroke Center of Linyi People’s Hospital affiliated with Shandong University from February 2016 to September 2018. These patients were screened, and 71 patients were recruited in this study, of which 11 patients had HT at the time of re-examination, and the other 60 patients did not. Figure 1 shows the flowchart of the exclusion and inclusion of the patients in our study. Table 1 summarizes the clinical characteristics of the HT and No-HT cohorts.

The exclusion criteria of the patients were as follows: (1) the patient could not cooperate to complete the examination, or the data of the patient was incomplete; (2) the patient had psychological or mental illness; or the patient had a history of stroke and was under treatment; and (3) patients with severe liver, kidney and heart function diseases, or patients with a history of brain trauma, stroke, intracranial artery occlusion, intracranial hemorrhage, tumor, etc.

The inclusion criteria for patients were as follows: (1) the basic clinical information of the patient included age, gender, and medical history such as hypertension, heart disease, or diabetes; (2) the age was between 18 and 85 years old; (3) the definite time of onset of the cerebral infarction was provided, and the time from onset to imaging examination was within 24 h; (4) the NIHSS score of the patient at admission was greater than or equal to 4 (indicating obvious neurological dysfunction); (5) the imaging sequences of preliminary examination included T1-weighted, perfusion-weighted, diffusion-weighted MRI (i. e., T1w, PWI, DWI, respectively), and CT or SWI for excluding external cerebral hemorrhage; (6) re-examination of CT or SWI within 1–3 days to confirm whether there was transformation of cerebral hemorrhage.

### 2.2. MR Imaging

The MRI images of initial examination and re-examination were collected by Siemens 3.0T MR imaging system (MAGNETOM Verio, Siemens Medical Solutions, Germany) using an 8-channel head coil. The initial examination sequences included T1w, DWI, and PWI images. SWI or CT examination was also performed to identify hemorrhage. No intracranial hemorrhage was found in all patients during the initial examination. All patients were treated with intravenous thrombolysis and re-examined within 1–3 days. A total of 11 patients found the transformation of a cerebral hemorrhage on SWI or CT. For the two examinations, the imaging parameters of MRI were the same. The T1w MR images were acquired with the parameters as follows: repetition time: 1400 ms; echo time: 9 ms; slice thickness: 5 mm; voxel size: 0.7 mm × 0.7 mm × 6.5 mm. The PWI images were scanned with the parameters as follows: repetition time: 1500 ms; echo time: 30 ms; slice thickness: 4 mm; voxel size: 1.8 mm × 1.8 mm × 4.0 mm. The DWI images were scanned with the parameters as follows: b=0 or $b=1000\;s/{\mathrm{mm}}^{2}$; repetition time: 3300 ms; echo time: 105 ms; slice thickness: 4 mm; Voxel size: 1.6 mm × 1.6 mm × 4.0 mm.

Once the PWI and DWI images were scanned, the Siemens Mr-syngp medical image post-processing workstation was used to automatically generate the multiparametric MRI images, such as ADC image from DWI image, and CBF, CBV, MTT and TTP images from PWI image, as shown in Figure 2.

For each patient’s MRI multiparametric sequence, the image was first converted from DICOM format to NIfTI format with the help of MRICron software (https://www.nitrc.org accessed on 10 August 2018). Then, DWI and PWI images were registered to the space of the T1w image [26], and the resulting transformation matrix was applied to the ADC, CBF, CBV, MTT and TTP images, respectively, so that all sequences were registered to T1w image space.

The DWI and PWI images of all patients were manually labeled by a doctor with more than 10 years of radiation medicine experience to obtain the core infarct area, low blood flow abnormal perfusion area and normal brain area. The high signal area in the DWI image (b = 1000) is the core infarcted area. For PWI images, the areas where the gray level decreased compared with the contralateral side in CBF and CBV images, and the areas with increased gray level compared with the contralateral sides in MTT and TTP images were regarded as low blood flow abnormal perfusion areas. When the diagnosis was in disagreement, it is judged by the imaging diagnosis report issued by a junior doctor and a senior physician in the hospital.

## 3. Methods

The framework of our method is shown in Figure 3. First, the abnormal ROIs were delineated on each MRI sequence and their contralateral normal regions were found with the help of symmetrical brain structures automatically, as in this way, we obtained all ROIs. Besides, we also collected 16 clinical factors. Second, the radiomics features from abnormal ROIs (pink arrow and bar) or normal ROIs (green arrow and bar), were extracted from five parametric MRI images, respectively. Then, the least absolute shrinkage and selection operator (LASSO) regression algorithm was applied to select two types of features and construct an Abnormal-ROIs model, an All-ROIs model and a clinical model. In addition, the statistical analysis on single feature prediction was used to screen the most distinctive clinical factors (brown arrow and bar) and integrate them with radiomics features, to develop a combined model. Below, the proposed method is described in detail from the feature extraction, feature selection and model construction of multiregional and multiparametric MRI images.

### 3.1. Image Preprocessing and ROI Segmentation

As shown in Figure 4, for T1w MRI images, we firstly used FreeSurfer [27] to automatically segment the brain region, thus obtaining a brain template with multiple subregions. Then, according to the clinical prior knowledge of the blood supply area of the cerebral artery on the abnormal side, the segmented brain regions were merged into 10 blood supply areas: anterior cerebral (a); middle cerebral artery (m1–m5); posterior cerebral artery (p); lenticular nucleus (l); caudate nucleus (c); and insula (i).

The infarct area, low blood flow perfusion area and normal brain area manually labeled by experts were intersected with the above 10 areas on the abnormal side, respectively, which will generate 10 ROIs on the abnormal side. According to the symmetry of the brain, the ROIs of the normal contralateral side were generated according to the combination of the brain regions corresponding to that from the abnormal side. Then, 10 normal ROIs were also obtained. Therefore, there were a total of 20 ROIs in each modal image as shown in Figure 4.

### 3.2. Multiregional and Multiparametric MRI Radiomics Feature Extraction

Based on the ROIs, we extracted three groups of features: (1) first-order features; (2) geometry features; and (3) texture features of the gray-level co-occurrence matrix. The features were extracted from images of five modalities within 20 labeled subregions, including 10 abnormal ROIs and 10 normal ROIs.

The extracted features were summarized in Table 2. Specifically, 18 intensity features described the first-order distribution of the multiregional intensities. Fourteen geometry features depicted the characteristics of the ROI shape. What’s more, 22 texture features were also extracted. Finally, for each patient, 5400 quantitative features ($20\times \left(18+14+22\right)\times 5=5400$) were extracted using PyRadiomics [28].

### 3.3. Feature Selection

As mentioned above, in the process of building the radiomics HT prediction model based on the ROI characteristics of the abnormal side, we extracted 2700 features for each sample, and the normal-ROIs and abnormal-ROIs features were concatenated for supplementary information of the normal side, so that the number of each sample’s feature is 5400 in total.

With high-dimensional multiregional features, we aimed to develop a reliable model with a minimal set of relevant features. Thus, LASSO [25] was used to select all-relevant features to build a reliable classification model. LASSO is the abbreviation of least absolute shrinkage and selection operator. LASSO is a linear regression method using L1 regularization to make some learned feature weights zero, so as to achieve the purpose of feature selection. Finally, for the construction of the All-ROIs HT prediction model and Abnormal-ROIs HT prediction model, we select 14 features with high lasso weight coefficients, respectively.

In addition, we verified the classification ability of 16 clinical features through statistical analysis combined with single feature prediction evaluation, to screen the clinical features that have strong support. Specifically, we used SPSS software (version 20, Chicago, IL, USA) to statistically evaluate the correlation between 16 clinical features and the dependent variable of HT. On the other hand, HT prediction models were established by using single clinical features and their classification performance was tested. Finally, the AUC value in the classification results is used to evaluate its classification ability, that is, the higher the AUC value, the better the classification performance of this feature. Finally, through the unified analysis of the above results and the prediction results, two clinical features with the best classification performance were selected. (SVS_ 1 and M2).

### 3.4. Data Split and Predictive Models Construction

The flow of dataset division and model training is shown in Figure 5. A total of 71 patients were collected in this study, of which 11 patients had HT at the time of re-examination, and the other 60 patients did not. All patients were blinded to the clinical and outcome information and were randomly allocated into a primary training cohort (*n* = 49) and a validation cohort (*n* = 22).

It is important to note that, since the proportion of positive samples is too small and there is a category imbalance in the dataset, the SMOTE [29] algorithm is used to generate additional positive samples in the primary cohort, to help the training of the model. In addition, due to the small amount of data, the method of five-fold cross-validation is used for model training, to obtain five classification results, and take the average value of them as the classification result of the model in the whole primary cohort. Finally, the trained model is tested on the independent validation cohort to get the final classification results.

We used RF to build four predictive models. Specifically, based on the selected 14 Abnormal-ROIs features, 14 All-ROIs features and 16 clinical factors, the Abnormal-ROIs model, All-ROIs model and clinical model were established by using RF, respectively. Besides, the filtered two clinical features were fused with 14 radiomics features from all ROIs, and LASSO regression was used again to realize the second feature selection. Finally, a similar model construction method is further used to establish the combined model.

### 3.5. Model Validation Metrics

All predictive models were trained on the primary cohort and tested on the independent validation cohort. The performance was assessed using accuracy (ACC), sensitivity (SEN), specificity (SPEC), area under the receiver operating characteristic (ROC) curve (AUC) and F1 Score.

## 4. Results

We implemented our method in Python and Scikit-Learn with Intel (R) Core (TM) i5–9600 K CPU and 64 GB of memory using the Linux system. All models were trained using the following hyperparameters: alpha, the regularization coefficient of LASSO of 1 × 10^−6^. While *n*_estimators (number of decision trees) and max_depth (max depth of decision trees) were set to seventy and five, respectively. Finally, the average time for training and predicting of all models can reach 2.5 s and 0.2 s, respectively.

### 4.1. Feature Selection

We used LASSO regression to reduce the 5400 features extracted from all ROIs to 14 most distinctive features according to the LASSO coefficient in descending order. The selected features consist of ten from abnormal ROIs and four from the normal side in CBF, CBV and ADC images, and these features are the first-order feature and shape feature calculated from anterior cerebral (a), middle cerebral artery (m1–m5), lenticular nucleus (l) and insula (i). The detail is shown in Figure 6.

It is easy to find that the first-order features have the strongest classification performance compared with the geometry and texture features. For example, the feature from the anterior cerebral of the abnormal side (e.g., ’CBF_a_abnormal_shape_MajorAxisLength’) has the highest coefficients, while the features from middle part of the abnormal side account for a larger proportion, which reflects the size and shape of the infarcted region, so they can be considered as features of the more important ROIs. Besides, it can be found that there are great differences in the contribution of different MRI sequences to the HT prediction, among which ADC has the strongest classification performance. More importantly, we find that there are four features extracted from normal ROIs in the 14 selected features, that is, even if there are no lesions, the normal ROIs also provide comparative information between normal and diseased brain tissue. Therefore, they can provide supplementary information for the prediction of HT.

To further confirm this conclusion, we remove the normal-ROIs features to focus on the abnormal ROIs, and finally select 14 features, as shown in Figure 7. These features are the ones with the largest LASSO coefficient from the original abnormal ROIs in ADC, CBF, CBV and MTT images, and these features are the first-order and shape features calculated from anterior cerebral (a), middle cerebral artery (m1–m4) and caudate nucleus (c).

In addition, we used statistical analysis and single feature prediction to evaluate the HT prediction ability for all 16 clinical features, and the results are shown in Table 1. It is easy to conclude that only SVS_ 1 and M2 are significantly correlated with HT (p_svs_1_ = 0.009, p_M2_ = 0.017). When we use each clinical feature to establish the prediction model separately, these two features also achieve the best results (AUC_svs_1_ = 0.692, AUC_M2_ = 0.610). Therefore, we believe that SVS_ 1 and M2 have a strong ability to predict HT.

Therefore, we combined them with 14 radiomics features from All-ROIs, and used the LASSO regression again to obtain 11 selected features, as shown in Figure 8. It can be seen that, after adding clinical information, the features from the ADC sequence still account for the vast majority of the total number of features. In addition, compared with other features, clinical feature M2 has the highest weight coefficient, that is, this feature plays an important role in the HT prediction.

### 4.2. Model Validation

Five-fold cross-validation is used for model training, and five models are obtained. Besides, to evaluate the prediction performance of the five models, their classification performance is tested on the validation cohort, respectively. The results are shown in Table 3.

It can be seen that the classification accuracy is low (AUC = 0.556) when only clinical features were used to predict HT, which means that the prediction results tend to have a large deviation.

Besides, the classification performance has been greatly improved when using radiomics features for prediction. In addition, compared with the radiomics model using only the information from the abnormal ROIs, the prediction performance after adding normal-ROIs features are better (AUC = 0.871). Therefore, we can further infer that these ROIs without lesions provide comparative information between normal and diseased brain tissue, so they have a certain information supplement for HT prediction.

Finally, when fusing the clinical factors with the radiomics features, the combined model achieves the best classification performance (AUC = 0.911). Therefore, we believe that although the prediction performance of the clinical model is poor, the clinical information also contains important information with strong classification ability. By searching these distinctive features and adding them to the model as supplementary information, the classification performance of the model can be significantly improved.

In addition, to verify the robustness of our method, we set three random seeds (100, 200, and 300), and plot the ROC curves of different models under different hyperparameters, as shown in Figure 9. It can be seen that different hyperparameters have little impact on the performance of the model and all three models can predict HT with AUC>0.82. What’s more, to further test the generalization of our method, we tested the model’s accuracy under different hyperparameters (Appendix A in the Appendix A). The results show that our method is insensitive to the changes of various hyperparameters and has good robustness.

## 5. Ablation Study

### 5.1. Prediction Models Based on Single-Sequence MRI Images

In order to fairly compare the HT prediction ability of each sequence image, we separately extracted and selected features from five single-sequence MRI images: ADC, CBF, CBV, MTT and TTP, further built five single-sequence HT predictive models.

Specifically, 54 features were extracted from each of the 20 ROIs of each sequence, and 1 × 20 × 54 = 1080 radiomics features were extracted from each single-sequence MRI image, and then LASSO regression was used to reduce the dimension of the features. The feature selection results of each sequence are shown in Figure 10.

We made further statistical analysis on the above-selected features and the results are shown in Figure 11. It can be found that for any sequence of ADC, CBF, CBV, MTT and TTP, the number of features from normal ROIs accounts for a large proportion. This further proves that even if there is no lesion in the normal ROIs, the information from normal ROIs still plays a key role in the task of HT prediction. As far as we know, this has not been found in previous HT prediction studies.

On the other hand, through the statistical results, we can easily find that the number of first-order features is greater than the number of shape features, and the number of shape features is much greater than the number of texture features. This shows that compared with texture features, first-order and shape features have stronger judgment ability for this classification problem.

Based on the above-selected features, the RF method was used to build five single-sequence models, and the classification performance of each model is evaluated on the independent validation cohort. The results are shown in Table 4. It can be seen that compared with the prediction results of the model based on multiparametric MRI images in Table 3, the prediction accuracy of the model using a single sequence is generally reduced. This shows that the different parametric MRI images have a certain contribution to the prediction of HT, which further proves the effectiveness of the multiparametric feature fusion method in the paper.

In addition, the classification accuracy of a single-sequence prediction model can be ranked as ADC > TTP >CBF >CBV > MTT. Among them, the ADC sequence still achieves the best prediction effect (AUC = 0.831) when the minimum number of features was selected. As mentioned earlier, most of the features in the radiomics feature selection results of multiparametric images are also from ADC images. It is further proved that the ADC sequence has good performance in the task of HT prediction.

### 5.2. The Number of Features

To evaluate the influence of the number of features on the classification accuracy of the HT prediction model, we selected different numbers of features from the 20 features with the highest LASSO coefficients for modeling and testing, the results are shown in Table 5.

It is easy to find that with the increase in the number of features, the classification accuracy of the model generally shows a trend of first increasing and then decreasing. Among them, when the number of characteristics is 14, AUC and ACC reach their peak. After that, although the AUC fluctuates to some extent, it shows a downward trend, and the decline of ACC is very obvious. This is due to the overlearning of the model when the number of features is too large, that is, the model performance is not robust and results in poor performance in the independent validation cohort.

## 6. Discussion

Intravenous thrombolysis is a common treatment for patients with acute ischemic stroke. However, intravenous thrombolytic therapy may cause further hemorrhage and worsen the patient’s condition. To help doctors to diagnose potential HT patients early and make a correct judgment on whether to carry out intravenous thrombolysis therapy, this study proposed a predictive model for HT prediction based on multiparametric MRI radiomics features and machine learning.

### 6.1. Predictive Model Performance

Four different predictive models were developed on a primary training cohort and validated using an independent dataset in the study. Particularly, the combined model including both clinical and radiomics features can predict the HT of patients with acute ischemic stroke with the performance of AUC = 0.911, which shows that our model has a discriminative ability for HT prediction. In addition, the predictive model just based on radiomics features achieves the classification accuracy of AUC = 0.871, and more importantly, when we remove the features from the normal ROIs and only focus on the information from the abnormal area, the classification accuracy of the prediction model decreases significantly (AUC = 0.831). 

We also conducted some ablation experiments. Firstly, we establish five single-sequence predictive models. The results show that the prediction accuracy of the single-sequence models is generally lower than that of our multiparametric model. Among them, the ADC sequence still achieves the best accuracy when the minimum number of features is selected. In addition, we select different numbers of features to establish a multiparametric predictive model. The results show that when the number of features is 14, the model has the best prediction performance. When more features are selected, the model appears to overfit.

What’s more, to verify the robustness of our method, we tested the prediction performance of the model under different hyperparameters. The results show that our method is insensitive to hyperparameters and has a good generalization ability.

### 6.2. Our Findings

Firstly, as shown in Figure 6 and Figure 11, the proposed method can be used as a powerful tool for HT prediction. Secondly, we found that among the final selected features, the features from the ADC sequence are far more than other sequences, and the prediction performance of the ADC model is significantly better than other single-sequence models, which shows that the ADC sequence is very important for HT prediction. Finally, we found that the features from normal ROIs accounted for a large proportion of the final selected features, whether for the multiparametric models or single-sequence models. Furthermore, when we removed the features from the normal ROIs and constructed an Abnormal-ROIs model, the prediction accuracy was significantly reduced. This shows that even if there is no lesion in the normal ROIs, it also provides comparative information between normal and diseased brain tissue, so it has a certain supplement effect on the prediction of HT. To the best of our knowledge, this is a finding that has not been found in previous studies. The possible reason may be that in previous studies, most of the patients with acute ischemic stroke were over 65 years of age, and the patients had undergone other changes such as ageing in the cerebral blood vessels and brain tissues [30]. Therefore, this affected the comparison of the brain tissue in the normal side to the brain tissue in the abnormal side. However, in recent years the stroke patients are getting younger [31], and the normal side of the brain of younger patients can provide good comparative information and is therefore a good addition to the information available for HT prediction.

### 6.3. Clinical Implications

At present, the most effective treatment for acute ischemic stroke (AIS) is completing recanalization using intravenous thrombolysis within the time window, and the success rate of treatment is closely related to time. However, due to the risk of HT, doctors cannot make decisions quickly.

Traditionally, doctors use clinical information to assess the risk of HT, but the method has limited accuracy and cannot well guide the use of intravenous thrombolysis (AUC = 0.556 in Table 3). As such, it is usually necessary to collect multiparametric MRI images of patients. Previous studies failed to fully utilize the MRI images, resulting in low prediction accuracy [21,22,32]. Besides, although deep learning has become a powerful tool for classification tasks in recent years, its training process often requires a large amount of data, which is often difficult to meet [23,24]. In addition, the poor interpretability of deep learning limits its wide application in clinical practice.

By extracting the radiomics features of multiparametric MRI images and fusing them with clinical factors, our method achieves good prediction results (AUC = 0.911). This proves that our method can make timely HT predictions concisely and efficiently.

### 6.4. Limitations and Further Research

There are some limitations to our study. First, The HT prediction model is constructed based on the multiparametric MRI sequence images of an institution. Therefore, the validity and generalizability of our findings await further investigation. In the future, multi-center clinical data sets will be used to further evaluate the performance of the prediction model. Second, the process of image preprocessing, ROI segmentation and feature extraction is complicated, and radiomics features are usually a shallow first-order feature, geometric feature and texture feature, whose representational power is inadequate. Specifically, extracting radiomics features requires clinical knowledge, which consumes a lot of human resources. Besides, such manual features only focus on the underlying image information such as shape, intensity and texture of the ROI and do not contain high-level semantic information. In future research, we will try to use deep learning methods to extract deeper potential features with a view to achieving better prediction results. Moreover, it is worth noting that the pathophysiology, prognosis and clinical features of acute small-vessel ischemic strokes are different from other acute cerebral infarcts [33]. Therefore, the results obtained in the study could only be extrapolated to non-lacunar acute cerebral infarcts.

In addition, several evidences suggest that cerebral atrophy is another potentially relevant manifestation of acute stroke related to the outcome in elderly individuals [34,35]. As such, apart from looking for better predictive accuracy, our future research will study the relationship between the impact of cerebral atrophy and prediction of HT in acute ischemic strokes with intravenous thrombolysis. we will also demonstrate the early predictive ability of the method in stroke-related complications such as epilepsy.

## 7. Conclusions

In this study, our method deeply mined the radiomics features of multiparametric MRI images and fused them with clinical factors, showing sufficiently high discrimination for HT of patients with acute ischemic stroke. Specifically, four predictive models were developed and validated. The Radiomics features were extracted from five MRI sequences, and the dimensions of features were reduced by LASSO regression, thus an HT prediction model based on radiomics features was constructed. In addition, the clinical features with strong prediction ability were screened out by the combination of statistical analysis and single feature prediction, and fused with radiomics features, and the HT prediction model based on fusion features was further constructed by the RF method. The model is simple and efficient, so it may be clinically useful. In addition, we found that the contralateral region of the lesion area plays an important role in the prediction of HT. As far as we know, this has not been found in previous studies. 

## Figures and Tables

**Figure 1 brainsci-12-00858-f001:**
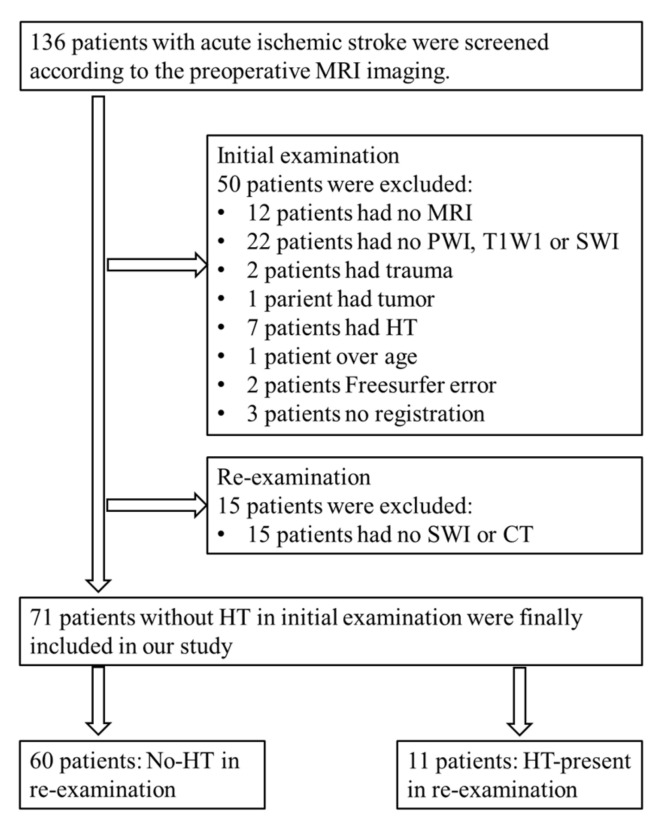
Flowchart of exclusion and inclusion of patients in our study.

**Figure 2 brainsci-12-00858-f002:**
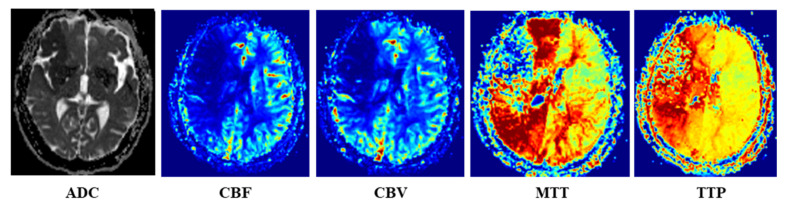
Multiparametric MRI images (ADC from DWI, and CBF, CBV, MTT and TTP images from PWI image).

**Figure 3 brainsci-12-00858-f003:**
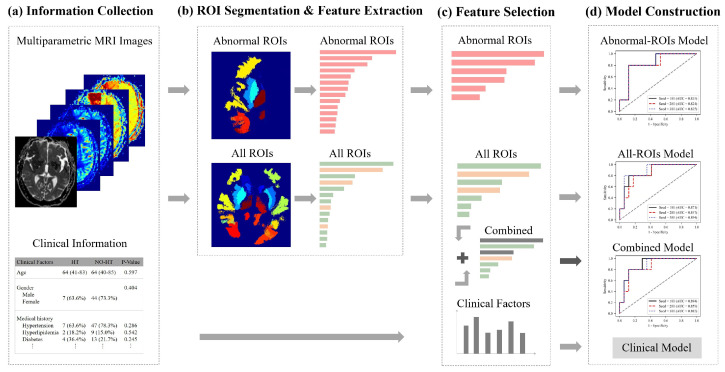
The framework of our method.

**Figure 4 brainsci-12-00858-f004:**
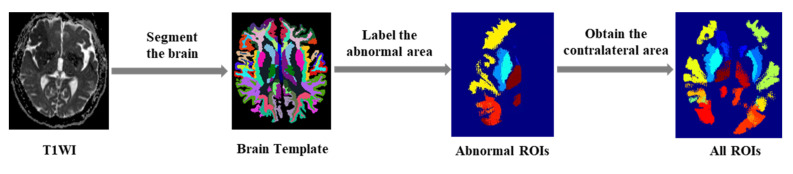
The process of the ROIs segmentation.

**Figure 5 brainsci-12-00858-f005:**
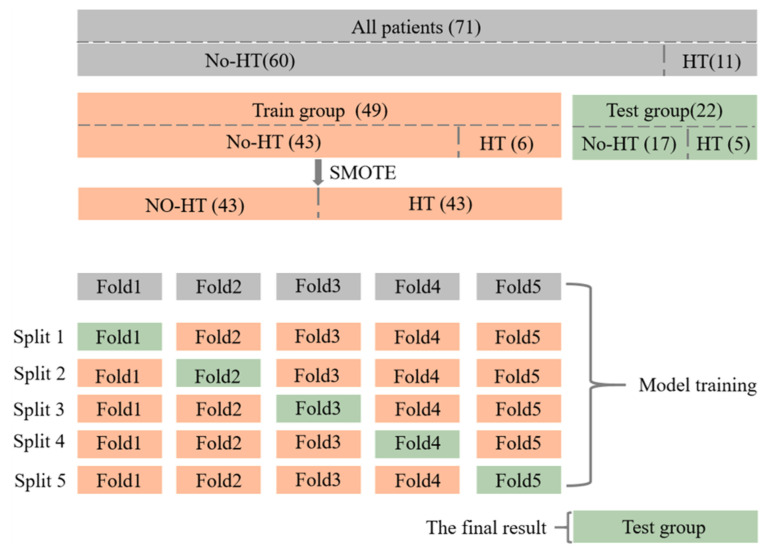
The process of data split and model training.

**Figure 6 brainsci-12-00858-f006:**
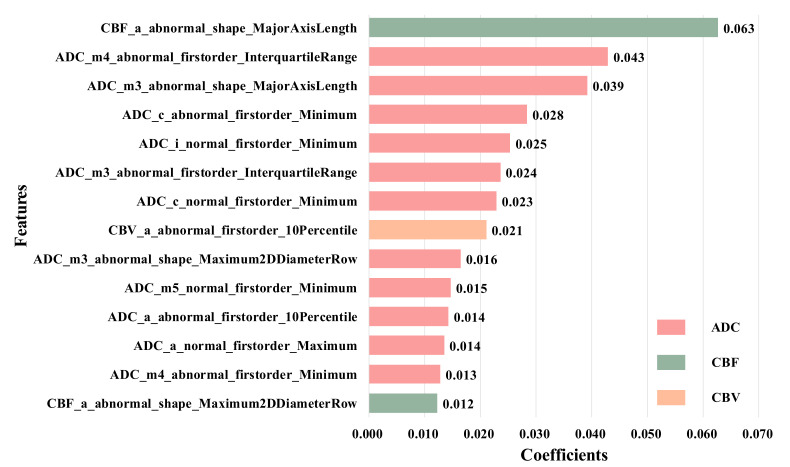
Features extracted from all ROIs. The naming format of features is: Categories of MRI images_Name of ROI_Categories of ROI_Name of feature. The feature naming method in the following feature selection diagram is the same as this.

**Figure 7 brainsci-12-00858-f007:**
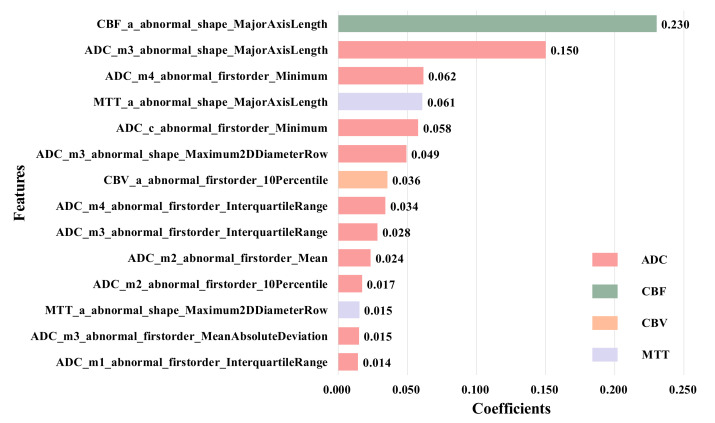
Features extracted from the abnormal ROIs.

**Figure 8 brainsci-12-00858-f008:**
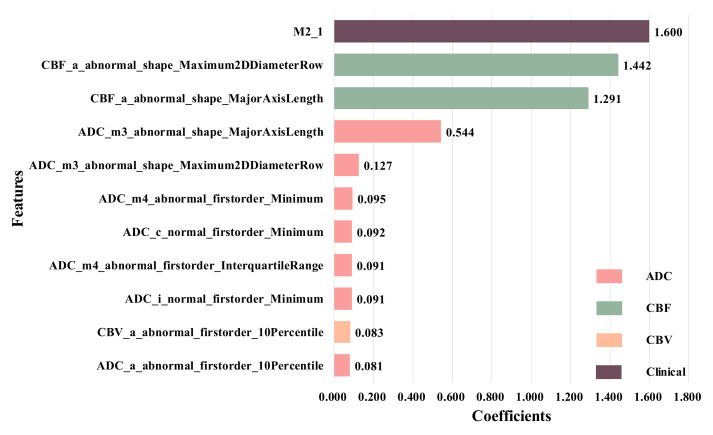
Extracted fusion features.

**Figure 9 brainsci-12-00858-f009:**
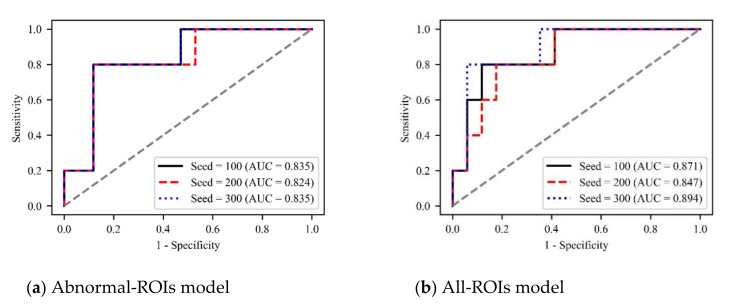
Illustration of the ROC curves based on Abnormal-ROIs model (**a**), All-ROIs model (**b**) and Combined model (**c**), respectively.

**Figure 10 brainsci-12-00858-f010:**
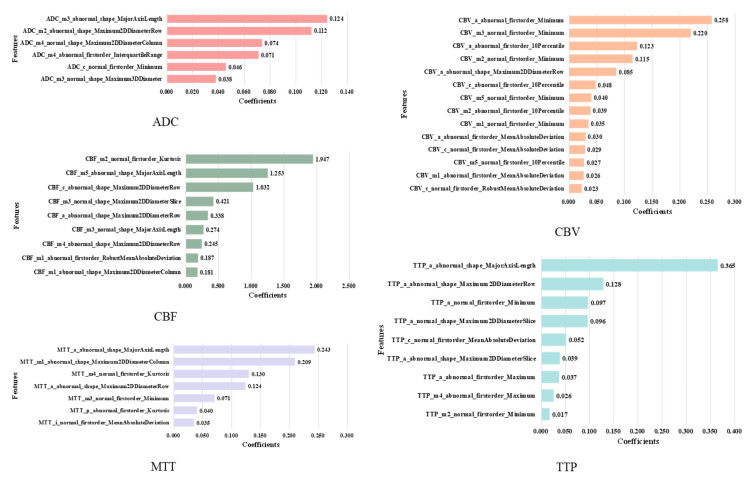
Features extracted from 5 single-sequence images.

**Figure 11 brainsci-12-00858-f011:**
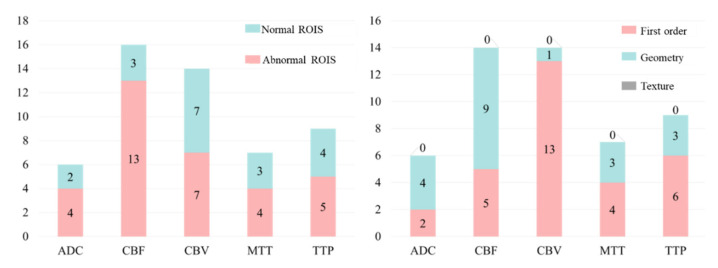
Analysis of features obtained from single-sequence images.

**Table 1 brainsci-12-00858-t001:** Clinical information and characteristics in the HT and No-HT groups of patients.

Clinical Characteristics	HT (*n* = 11)	No-HT (*n* = 60)	*p*-Value	AUC
Age, median (Range)	64 (41–83)	64 (40–85)	0.597	0.524
Gender			0.404	0.540
Male	7 (63.6%)	44 (73.3%)		
Female	4 (36.4%)	16 (26.7%)		
Medical history				
Hypertension	7 (63.6%)	47 (78.3%)	0.286	0.565
Hyperlipidemia	2 (18.2%)	9 (15.0%)	0.542	0.516
Diabetes	4 (36.4%)	13 (21.7%)	0.245	0.579
Atrial fibrillation	2 (18.2%)	3 (5.0%)	0.169	0.566
Leukoaraiosis	7 (63.6%)	40 (66.7%)	0.549	0.535
Coronary	2 (18.2%)	3 (13.3%)	0.485	0.524
Infarct location				
A	1 (9.1%)	6 (10.0%)	0.705	0.505
M1	6 (54.5%)	32 (53.3%)	0.602	0.506
M2	4 (36.4%)	35 (58.3%)	0.017	0.610
P	1 (9.1%)	6 (10.0%)	0.650	0.504
Clinical score				
SVS_1	11 (100.0%)	37 (61.7%)	0.009	0.692
SVS_2	3 (27.3%)	22 (36.7%)	0.408	0.547
mTICI_2	5 (45.5%)	28 (46.7%)	0.155	0.539
Recanalization state			0.169	0.511
yes	5 (45.5%)	26 (43.3%)		
no	6 (54.5%)	34 (56.7%)		

Note: Data are presented as *n* (%). A = anterior cerebral; M1, M2 = middle cerebral artery; *p* = posterior cerebral artery; SVS_1 = susceptibility vessel sign at initial examination; SVS_2 = susceptibility vessel sign at re-examination; mTICI_2 = modified treatment in cerebral ischemia score at re-examination.

**Table 2 brainsci-12-00858-t002:** A summary of the high-throughput radiomics features extracted.

Feature Classes	Feature Names
First-order Features	10Percentile, 90Percentile, Energy, Entropy, Interquartile Range, Kurtosis, Maximum, Mean Absolute Deviation, Mean, Median, Minimum, Range, Robust Mean Absolute Deviation, Root Mean Squared, Skewness, Total Energy, Uniformity, Variance
Geometry Features	Elongation, Flatness, Least Axis Length, Major Axis Length, Maximum2DDiameterColumn, Maximum2DDiameterRow, Maximum2DDiameterSlice, Maximum3DDiameter, Mesh Volume, Minor Axis Length, Sphericity, Surface Area, Surface Volume Ratio, Voxel Volume
Texture Features	Autocorrela, Joint Average, Cluster Prominence, Cluster Shade, Cluster Tendency, Contrast, Correlation, Difference Average, Difference Entropy, Difference Variance, Joint Energy, Joint Entropy, Imc1, Imc2, Idm, Idmn, Id, Idn, Inverse Variance, Maximum Probability, Sum Entropy, Sum Squares

**Table 3 brainsci-12-00858-t003:** Classification results of all prediction models.

Models	AUC	ACC	SEN	SPEC	F1 Score
Clinical model	0.556 ± 0.045	0.545 ± 0.064	0.333 ± 0.471	0.556 ± 0.045	0.067 ± 0.094
Radiomics model					
Abnormal ROIs	0.831 ± 0.006	0.818 ± 0.000	0.600 ± 0.000	0.882 ± 0.000	0.600 ± 0.000
All ROIs	0.871 ± 0.019	0.848 ± 0.021	0.733 ± 0.094	0.882 ± 0.048	0.687 ± 0.029
Combined model	0.911 ± 0.009	0.894 ± 0.021	0.810 ± 0.067	0.933 ± 0.054	0.830 ± 0.023

**Table 4 brainsci-12-00858-t004:** Classification results of models based on single-sequence images.

Sequence (Number_Features)	AUC	ACC	SEN	SPEC	F1 Score
ADC (6)	0.831 ± 0.015	0.848 ± 0.021	0.600 ± 0.000	0.922 ± 0.028	0.644 ± 0.031
CBF (9)	0.769 ± 0.024	0.848 ± 0.021	0.533 ± 0.094	0.941 ± 0.000	0.611 ± 0.079
CBV (14)	0.733 ± 0.011	0.773 ± 0.000	0.000 ± 0.000	1.000 ± 0.000	0.000 ± 0.000
MTT (7)	0.694 ± 0.017	0.621 ± 0.021	0.933 ± 0.094	0.529 ± 0.000	0.527 ± 0.040
TTP (9)	0.780 ± 0.024	0.788 ± 0.021	0.667 ± 0.094	0.824 ± 0.048	0.587 ± 0.030

**Table 5 brainsci-12-00858-t005:** Classification accuracy of models with different number of features.

Number_Features	AUC	ACC	SEN	SPEC	F1 Score
5	0.765 ± 0.017	0.712 ± 0.057	0.667 ± 0.249	0.725 ± 0.139	0.500 ± 0.071
6	0.745 ± 0.006	0.712 ± 0.113	0.733 ± 0.249	0.706 ± 0.220	0.544 ± 0.020
7	0.784 ± 0.015	0.773 ± 0.098	0.467 ± 0.249	0.863 ± 0.194	0.468 ± 0.100
8	0.800 ± 0.010	0.818 ± 0.037	0.467 ± 0.094	0.922 ± 0.055	0.539 ± 0.068
9	0.831 ± 0.006	0.788 ± 0.021	0.800 ± 0.000	0.784 ± 0.028	0.632 ± 0.024
10	0.843 ± 0.006	0.818 ± 0.037	0.600 ± 0.163	0.882 ± 0.083	0.594 ± 0.070
11	0.847 ± 0.017	0.773 ± 0.037	0.600 ± 0.000	0.824 ± 0.048	0.548 ± 0.041
12	0.835 ± 0.017	0.758 ± 0.021	0.800 ± 0.000	0.745 ± 0.028	0.601 ± 0.021
13	0.859 ± 0.010	0.848 ± 0.021	0.733 ± 0.094	0.882 ± 0.000	0.685 ± 0.060
14	0.871 ± 0.019	0.848 ± 0.021	0.733 ± 0.094	0.882 ± 0.048	0.687 ± 0.029
15	0.859 ± 0.025	0.818 ± 0.037	0.667 ± 0.094	0.863 ± 0.055	0.626 ± 0.057
16	0.863 ± 0.015	0.803 ± 0.021	0.800 ± 0.163	0.804 ± 0.073	0.644 ± 0.031
17	0.859 ± 0.000	0.803 ± 0.021	0.667 ± 0.094	0.843 ± 0.028	0.604 ± 0.050
18	0.812 ± 0.000	0.788 ± 0.021	0.667 ± 0.094	0.824 ± 0.000	0.586 ± 0.057
19	0.847 ± 0.029	0.773 ± 0.037	0.800 ± 0.163	0.765 ± 0.083	0.612 ± 0.050
20	0.851 ± 0.006	0.712 ± 0.021	0.867 ± 0.094	0.667 ± 0.028	0.577 ± 0.038

## Data Availability

The data presented in this study are available on request from the corresponding author. The data are not publicly available due to privacy issues.

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
