# Peer review of "Prediction Model of Hemorrhage Transformation in Patient with Acute Ischemic Stroke Based on Multiparametric MRI Radiomics and Machine Learning"

_brainsci, 2022, doi:10.3390/brainsci12070858_

Round 1

Reviewer 1 Report

The work is very well edited, objective, appropriate figures help in understanding.

I have just one more important remark. Authors say at Discussion section: "Even if there is no lesion in the normal ROIs, it also provides comparative information between normal and diseased brain tissue, so it has a certain supplement effect on the  prediction of HT."

It is a very interesting statement and i think its worth to discuss it in detail.

What could be the likely mechanism for this?

Is it possible to distinguish between less relevant, informative ROIs and essential ones by radiological methods?

Author Response

Point 1: The work is very well edited, objective, appropriate figures help in understanding. I have just one more important remark. Authors say at Discussion section: "Even if there is no lesion in the normal ROIs, it also provides comparative information between normal and diseased brain tissue, so it has a certain supplement effect on the prediction of HT. "It is a very interesting statement and I think it’s worth to discuss it in detail. What could be the likely mechanism for this?

Response 1: We appreciate the reviewer’s insightful suggestion and agree that it would be useful to to explain the biological mechanism of the help of normal side to HT prediction.

Firstly, as shown in Figure 6 and 11, we found that the features from normal ROIs accounted for a large proportion of the final selected features, whether for the multiparametric models or single-sequence models. And when we removed the features from the normal ROIs and constructed an Abnormal-ROIs model, the prediction accuracy was significantly reduced. This show that even if there is no lesion in the normal ROIs, it also provides comparative information between normal and diseased brain tissue, so it has a certain supplement effect on the prediction of HT. To the best of our knowledge, this is a finding that has not been found in previous studies. The possible reason may be that in previous studies, most of the patients with acute ischemic stroke were over the age of 65 years, and the patients had undergone other changes such as ageing in the cerebral blood vessels and brain tissues. Therefore, this affected the comparison of the brain tissue in the normal side to the brain tissue in the abnormal side. However, in recent years the stroke patients are getting younger, and the normal side of the brain of younger patients can provide good comparative information, so they are the good addition to the information available for HT prediction. (lines 436 to 454 in the manuscript).

Point 2: Is it possible to distinguish between less relevant, informative ROIs and essential ones by radiological methods?

Response 2: We greatly appreciate this comment. In our manuscript, we used LASSO regression for feature selection. As shown in Figure 6, the final filtered features are ranked in descending order of coefficient, where a higher coefficient proves that the feature is more important for HT prediction. As shown in Figure 6, we can initially distinguish between less relevant, informative ROIs and essential ones for the final filtered features. For example, the feature from the anterior cerebral of the abnormal side (’CBF_a_abnormal_shape_MajorAxisLength’) has the highest coefficients, while the features from middle part of the abnormal side account for a larger proportion, which reflects the size and shape of the infarcted region, so they can be considered as the more important ROIs. (lines 290 to 294 in the manuscript).

Reviewer 2 Report

In their study, the authors proposed a predictive model of hemorrhagic transformation (HT) in a sample of patients with acute ischemic stroke with intravenous thrombolysis therapy based on multiparametric MRI radiomics and fused with clinical features  and machine learning. The authors developed and validated four different predictive models. They  concluded that combining clinical factors with radiomic features benefited the prediction performance (AUC=0.91, accuracy=0.894) of HT. The study is potentially interesting, but can be improved if the following considerations are addressed:

1.    Delete lines 30-39 of the Introduction and start the statement at line 40.

2.    The Discussion  should clearly state that the pathophysiology, prognosis and clinical features of acute small-vessel ischemic strokes are different from other acute cerebral infarcts (see and add this recent reference: Int J Mol Sci 2022; 23, 1497). Therefore, the results obtained in the study could only be extrapolated to non-lacunar acute cerebral infarcts

3.    It would be interesting if the authors included in the text some of the limitations of their study.

4.    It would be interesting to emphasize in the text that a future line of research on the discussed topic would be to study the relationship and relevance of the impact of cerebral atrophy and  prediction of HT in acute ischemic strokes with intravenous thrombolysis. This recommendation is because cerebral atrophy is another potentially relevant (although still poorly characterized) manifestation of acute stroke related to outcome in elderly individuals (see and add these references:  Cerebrovasc Dis. 2010;30(2):157-66; Neurology. 2012 Nov 13;79(20):2016-7).

Author Response

Point 1: Delete lines 30-39 of the Introduction and start the statement at line 40.

Response 1: Thanks for your comment. We are very sorry that we did not catch the error in lines 30-39 in the previous manuscript and have made changes in response to your suggestions.

Point 2: The Discussion should clearly state that the pathophysiology, prognosis and clinical features of acute small-vessel ischemic strokes are different from other acute cerebral infarcts (see and add this recent reference: Int J Mol Sci 2022; 23, 1497). Therefore, the results obtained in the study could only be extrapolated to non-lacunar acute cerebral infarcts.

Response 2: We greatly appreciate this comment. As you said, the pathophysiology, prognosis and clinical features of acute small-vessel ischemic strokes are different from other acute cerebral infarcts. We reviewed and added the reference you recommended (Int J Mol Sci 2022; 23, 1497), and clearly pointed out in 6.4 that the results obtained in this study could only be extrapolated to non-lacunar acute cerebral infarcts. (lines 482 to 486, lines 603 to 609 in the manuscript)

Point 3: It would be interesting if the authors included in the text some of the limitations of their study.

Response 3: Thanks for your comment. There are some limitations to our study. First, The HT prediction model was constructed based on the multiparametric MRI sequence images of an institution. Therefore, the validity and generalizability of our findings await further investigation. Second, the process of image preprocessing, ROI segmentation and feature extraction was complicated, and radiomics features are usually shallow first-order feature, geometric feature and texture feature, whose representational power is inadequate. In future research, we will try to use deep learning methods to extract deeper potential features with a view to achieving better prediction results. What’s more, as you said in Point 2, it is worth noting that the pathophysiology, prognosis and clinical features of acute small-vessel ischemic strokes are different from other acute cerebral infarcts. Therefore, the results obtained in the study could only be extrapolated to non-lacunar acute cerebral infarcts.

We discussed the above limitations in detail in section 6.4 of the revised manuscript (lines 470 to 492 in the manuscript).

Point 4: It would be interesting to emphasize in the text that a future line of research on the discussed topic would be to study the relationship and relevance of the impact of cerebral atrophy and prediction of HT in acute ischemic strokes with intravenous thrombolysis. This recommendation is because cerebral atrophy is another potentially relevant (although still poorly characterized) manifestation of acute stroke related to outcome in elderly individuals (see and add these references: Cerebrovasc Dis. 2010;30(2):157-66; Neurology. 2012 Nov 13;79(20):2016-7).

Response 4: We grately appreciate for your comment. After reviewed the reference you recommended, we found that cerebral atrophy is indeed another potentially relevant manifestation of acute stroke related to outcome in elderly individuals. We have therefore been inspired by your suggestion to try to study the relationship of the impact of cerebral atrophy and prediction of HT, and have detailed this future work in section 6.4 of the revised manuscript. (lines 470 to 492, lines 603 to 609 in the manuscript).